# The Social Ecology of Caregiving: Applying the Social–Ecological Model across the Life Course

**DOI:** 10.3390/ijerph21010119

**Published:** 2024-01-22

**Authors:** Maggie T. Ornstein, Christine C. Caruso

**Affiliations:** 1Psychology, Sarah Lawrence College, Bronxville, NY 10708, USA; 2Bailey College of the Environment, Wesleyan University, Middletown, CT 06459, USA; ccaruso@wesleyan.edu

**Keywords:** caregiving youth, youth caregivers, young caregivers, young adult caregivers, family caregiving, social–ecological model, public health, composite vignette

## Abstract

Family caregivers provide care to people with disabilities, as well as ill and older adults, often with little to no outside assistance from the formal long-term care system. They are the backbone of long-term care, and it is a misconception that the majority of people institutionalize disabled people and older adults in the United States. Youth caregiving is under-examined in the field of public health and is in need of theoretical and practical attention. Building upon the work of Talley and Crews and Bronfenbrenner, we aim to broaden the scope of the discussion around caregiving through the application of the social–ecological model (SEM) to inform research and practice. This paper picks up where they left off, digging deeper into the ecological model to reimagine research, policy, and practices related to youth and young adult caregivers that are rooted in this framework. This application highlights care as embedded in social relations while allowing for an exploration of the ways structural barriers impact the caring unit. Looking holistically at the unit, rather than individuals as service users, provides an opportunity for understanding the interconnectedness of those giving and receiving care. It does so by rendering visible the interdependence of the caring unit, and the myriad structures, which bear down on care at the individual and household levels. This approach runs counter to dominant thinking, which focuses exclusively on the individuals involved in caregiving relationships, rather than considering them as interdependent units of care. This paper provides an analytic contribution, utilizing a narrative composite vignette based on literature and previous research.

## 1. Introduction

### State of Caregiving in the US

The majority of long-term care in the US is provided at home, with those receiving care relying heavily on unpaid help from family and friends [1,2,3]. The Centers for Disease Control and Prevention (CDC) has recognized family caregivers as the “backbone of long-term care provided in people’s homes” [4]. Respite services are the primary form of formal in-home support for caregivers but have a utilization rate of only 14% [1,2]. This is an example of the failure of policies and interventions to effectively support caregivers and the need to re-examine current approaches to research, policy, and interventions for this population [5,6].

Care, both the need for it and the provision of it, requires increased attention and policy reform. There are an estimated 53 million family caregivers (18+) in the US contributing USD 600 billion worth of care annually [7]. Recent research estimates that there may be as many as 5.4 million unaccounted child caregivers in the US [2] and 3.6–5.5 million young adult caregivers, aged 18–25, who provide care to family members [8]. Importantly, the labor by these younger caregivers has not been adequately accounted for, despite estimates that the value of youth caregiving is USD 8.5 billion [9]. Youth and young adult caregivers require increased attention, as indicated by the conceptualization of the US as emerging in terms of awareness, interventions, and policy responses to caregiving youth, compared to other countries, such as the UK [10]. The need for a life course approach to caregiving is especially relevant for caregiving youth, given the period of life stage at which their caregiving begins. There is an estimated USD 324,000 loss of income and benefits over the lifetime of caregivers, with this figure increasing to USD 659,139 when childcare is factored in [6,11]. Research has not yet explored the extent of financial implications specifically for people who began caregiving in their youth.

Limited research exists on caregiving in the field of Public Health [8,12,13,14,15]. Where it exists, it focuses on either care receivers, paid caregivers [16,17,18,19], or family caregivers [2,20] in isolation, despite calls for more integration of care networks [21,22]. One notable exception involves the bringing together of the needs of care receivers with dementia and their family caregivers [23]; however, paid caregivers are not often integrated into this work [24]. Furthermore, research with paid caregivers often fails to account for family caregivers also involved in providing care [25]. In instances when they are integrated, the relationships are often seen as oppositional, rather than reciprocal [19,26]. 

While literature exists supporting our proposal of the need for an integrated [6,22,27] relational and ecological approach to care, it remains limited, especially in the domain of public health [28,29,30,31]. When caregivers are included, it is often in the context of how to continue to provide the best care for care receivers, rather than to ensure everyone’s needs are being met. The current approach creates silos that lead to policy and practice, failing to adequately meet the needs of any of these individuals. Moreover, work needs to be conducted to bring us up to the current moment, characterized by lessons learned and continued implications of the COVID-19 pandemic [32,33]. Given the multiple dimensions that influence the experiences and outcomes of both caregivers and care recipients, a model such as Bronfenbrenner’s ecological systems theory is well suited to enhancing the understanding and analysis of caregiving [34]. This is particularly relevant especially across the life course and in relation to how upstream social determinants of health influence individuals and families. 

## 2. Public Health and the Ecological Turn

More than twenty years ago, Tebb and Jivanjee, coming from the social welfare field, addressed caregiver isolation from an ecological perspective, remaining focused on the individual level [35]. Since then, little has been written about caregiving from this perspective, although we argue that the framework is useful for the development of services and policies, which impact caregivers and care receivers.

In 2007, Talley and Crews conceptualized caregiving as a public health issue to widen the scope of caregiving research and draw attention to the overlapping needs of paid and family caregivers and care receivers [15]. However, research, policy, and practice continue to be compartmentalized. Talley and Crews recognized the enmeshed nature of caregiving relationships, to recast caregiving in an ecological model. In doing so, they pointed to ways in which structures and demographic changes, such as improvements in medicine and technology, shortages of nurses and health care workers, the 1999 Olmstead decision [36], and increased involvement of women in the paid workforce, have created the health care system’s dependence on family caregivers, while offering little in return for their increasing and myriad responsibilities [37]. They highlighted that caregiving is a life course experience, each caregiving situation involves multiple health dynamics, and caregiver and care receiver health is inextricably bound. As a result, they proposed a systemic view within public health and developed a triadic model of caregiving, which includes factors influencing the caregiver, care receiver, and professional care team [15]. This model acknowledges the needs and strengths of each member, as well as the relationships among them, and the impact on the caregiving experience. They argued that an increased understanding of caregiving has not translated into progress in caregiving policy or practice and that caregivers deserve more attention from the public health community. In response to the work of Talley and Crews [15], Eckenwiler makes a call for an explicit ecological approach to caregiving [12]. Building on the work of these researchers, we respond to this call for an ecological approach to propose research, policies, and practices that are rooted in the ecological model.

The coronavirus pandemic has highlighted the deep inequities around caregiving, which persist and have been exacerbated by the global pandemic. Research suggests a clear disparity in the negative impacts brought on by the pandemic, with family caregivers experiencing more negative outcomes, as compared with non-caregivers [38,39]. Additionally, it has been noted that attention needs to be paid to a new generation of caregiving youth as a result of family members who are now disabled due to long-COVID [33]. The pandemic emphasized the enmeshed nature of caregiving and, therefore, the inadequacy of the current care infrastructure. However, few policies support caregivers, in general, with even fewer that integrate the needs of caregiving youth. 

## 3. Ecological Systems Theory

Building on the work of Talley and Crews and Bronfenbrenner, we explicitly apply the social–ecological model (SEM) to inform research and practice against the backdrop of the COVID-19 pandemic [15,34]. Bronfenbrenner’s ecological systems theory is rooted in developmental psychology and describes how individuals are linked to dynamic social systems across the life course. Bronfenbrenner hypothesized that in addition to factors associated with individuals, such as age, sex, and health status, human behavior and development are impacted by five environmental systems: the micro, meso, exo, macro, and chronosystems, as illustrated in Figure 1. 

At each of these levels, there are opportunities for intervention. In this theory, the individual is at the center, nested within the microsystem which includes family, friends, coworkers, and peers. The Mesosystem is the neighborhood and institutional level and the physical environment, broadly conceptualized. The exosystem is made up of the economic, political, and educational systems. The macrosystem is the set of overarching beliefs and values, such as cultural and political ideologies (e.g., gender norms). Lastly, the chronosystem includes socio-historical conditions and patterns of events and transitions over a life course and takes into account changes over time. For example, the ways in which being a youth caregiver sets the stage for the ways individuals will experience and engage with subsequent events for the rest of their lives.

Bronfenbrenner’s model is widely cited in community health literature, with a variety of health promotion efforts, for example, with interventions related to obesity and food deserts [40,41], healthy environments [42,43], drug use [44], and school engagement, and to increase understanding of health disparities [45,46]. This model has been identified as useful for research addressing caregiving but has not yet been applied [12,15,35]. In general, weaknesses of this model include an oversimplification of system levels, a continued focus on proximal influences, and a limited use of the chronosystem, which examines changes over time [47]. 

## 4. Methods

We created a composite vignette to illustrate the complexity of caregiving relationships and conditions [48,49]. Vignettes are utilized to demonstrate the lived experiences of research participants to readers who may be unfamiliar with these firsthand accounts. Composite vignettes represent an integration of multiple participants’ stories into a single narrative [50]. An important ethical benefit is that it maintains the anonymity of participants. This vignette was constructed through the integration of aggregated data from the first author’s prior research [51]. We draw on this vignette to apply the ecological model and demonstrate the ways the current structure of long-term care and policies and interventions shape the conditions of caregiving and receiving care over the life course.

### Vignette-Caregiving in the Time of COVID

Tamara is middle-aged and has been caring for her father, Marvin, for nearly 30 years following a severe stroke he suffered while she was in high school. He was hospitalized for many months, with multiple transitions between acute, sub-acute, and long-term care facilities before returning home to the care of Tamara and her great aunt, Rosie. Home care services were supposed to begin immediately but never materialized. Eligibility for services was the first hurdle, with initial applications being denied without the family ever being informed. The family continued to assume services would begin and were only notified after months of unsuccessful attempts by Tamara to reach an administrator who could provide accurate information about Marvin’s case. This resulted in Tamara managing the care of both her father and great aunt while going to college and working. This precarious caregiving situation continued for 15 years, with home care services being so unreliable that there were long periods of time when no services were in place. While being a caregiver for her father, Tamara became a mother, and Rosie died at age 99. Without reliable care, Tamara was forced to forgo paid employment outside of the home for various periods of time.

Marvin, being low-income and uninsured, and having become disabled in his 40s, was able to obtain coverage through Medicaid (the health insurance program for people with limited income and resources) and Medicare (health insurance for people over 65, and some people with disabilities) immediately. Rosie owned the home they shared and had saved enough money throughout her life to build savings and have private insurance in addition to Medicare. The long-term care program Marvin was on provided a small monthly financial housing subsidy. Despite the privilege of homeownership, Tamara’s work constraints due to caregiving meant they did not have the financial resources to adequately maintain the household. This subsidy was instrumental in maintaining the household where intergenerational care was provided. Tamara applied for home care services after a bad fall left Rosie bedridden. Navigation of the long-term care system was a burden, which came with significant additional stress, as she provided 24-h care alone to multiple members of the household.

When services finally started, Audre, the home care worker they were assigned, was employed by an agency paid for by Marvin’s Medicaid. Tamara paid her privately to assist with Rosie’s care as well. If not for this extra pay, Audre would have had to reduce the time working with the family in order to obtain extra hours with other families. This extra financial burden is not without sacrifice for Tamara, but she is both in need of the help for Marvin and acutely aware of the difficulties Audre faces in being part of this low-paid, but essential workforce.

The years between Rosie’s death and the appearance of COVID-19 seemed to be a walk in the park compared to what pandemic life felt like for Tamara. Audre lives in an intergenerational household, which includes her elderly parents. Tamara’s household has also grown over the years, including an elderly cousin who came to live with them after a fall. Together, there are a dozen people to consider in an effort to protect against COVID-19. 

Tamara was lucky to work for a company, which allowed her to work remotely. Her husband’s job shut down early on, allowing the household the ability to isolate to protect themselves from COVID-19. Being in a surging city in California, the thought of home care continuing was terrifying, due to the risks for each household. Tamara and Audre discussed how to proceed during the early days of the pandemic. Together they decided to suspend services, leaving Tamara with both full-time work-from-home and full-time caregiving responsibilities. 

## 5. Caregiving and the Social–Ecological Model

An application of the SEM accounts for the complexity and bidirectionality of the levels and their impacts on one another. In applying an ecological model to the vignette, we acknowledge how these systems interact with each other. We propose a re-centering of care to provide a more inclusive framework and relational model, which includes all individuals involved in care relationships. Currently, there is an artificial separation of care receivers, family caregivers, and paid caregivers. Building off of Talley & Crews, we move to integrate the triad, which conventionally comprises the individual in the center [15]. The other individuals are typically located in the microsystem, but our application includes them in the unit of analysis, highlighting the need to integrate each of their needs and experiences simultaneously. This allows for the leveraging of each of the subsequent systems and social determinants of health in relation to research, practice, policy, and culture to promote the health of the unit, rather than each individual, which is the current focus in caregiver policy and service provision. 

In the vignette, Audre, Marvin, and Tamara would be at the center, rather than Marvin alone, with microsystems being the proximal environmental and social factors, which occur in face-to-face settings [34]. Recasting this configuration highlights the nuance and importance of the individuals immediately involved in care while acknowledging the ways in which the institutional role of the agency, which is located at the mesosystem level, impacts the microsystem, which is the household. Currently, the household microsystem is largely ignored, emphasizing the hidden impact of the mesosystem (agency policies and practices) and exosystem (government regulations) on household dynamics and functioning. In the chronosystem, responses to care would be life-course sensitive, and simultaneously responsive to the various life stages of Audre, Tamara, Rosie, and Marvin. In current practice, Tamara’s long caregiving career, spanning decades, is unaccounted for. In conventional analysis, Audre, the paid home care worker, is a representative of a service agency at the mesosystem level. 

Mesosystems represent the interactions and processes (relationships) that take place between two or more microsystems, containing the individuals [34]. In our example, Audre, as the paid caregiver coming into the home, becomes part of the microsystem, working together with Tamara and Rosie as part of the caring unit for Marvin, while simultaneously providing care to Rosie, along with Tamara. Typically, this integration is obscured when workers are conflated with home care agencies. Exosystems comprise the distal determinants and institutions, which shape the societal-level conditions that impact health [34]. Here, Audre is formally employed as Marvin’s caregiver through a community-based agency. 

The macrosystem consists of cultural ideologies, which form the characteristics of the micro, meso, and exosystems of a particular culture or subculture. The macrosystem in the United States is rooted in individualism [52] and tells us that each individual and family needs to figure out how to accomplish the caregiving needed, rather than looking toward policy changes that will impact the conditions of care. When caregiving is viewed through a lens of family, duty, and responsibility, negative effects related to the caregiving role diminish [53,54]. Applying the SEM to caring units emphasizes interdependence, rather than individualism and independence. This is especially relevant to caregiving youth, as there is little family-level data gathered in the research [54]. These macrosystem cultural norms lead to opportunities or a lack thereof and “life course options” that are deeply ingrained in these systems [34]. As a student at the onset of care, all of Tamara’s subsequent decision-making about her life was entwined with the cultural ideology of the individual and personal responsibility for family care. Her caring role constrained the choices she was able to make for herself, such as where to attend college so she could continue to live at home to provide care, the paid employment she engaged with, her personal relationships, and decisions about having children, all while balancing her caregiving responsibilities. The barriers to services in the exosystem exacerbated the difficulties in providing care.

The chronosystem deals with the “degree of hecticness/ability in daily life” and the passage of time, which includes changes over time [34], p. 6. Typically, these changes are applied to Marvin, as his conditions change with age. The exclusive focus on Marvin, as the care receiver, erases the impacts on his caregivers and the effects of the caregiving situation over time. Research related to the social support needs of caregiving youth over time is limited and in need of attention [55]. A focus on the chronosystem would provide a life course approach and respond to the call for longitudinal research to better understand long-term outcomes related to caregiving [56]. Furthermore, the exploration of social support for youth caregivers should span each level of the SEM to include analysis across systems, including home- and community-based support, social services, school, and medical professionals, as well as more distal and indirect, but essential sources of support.

## 6. Discussion

To illustrate this application of the SEM to caregiving youth, we provide two examples that apply this relational shift, which would have wide-reaching impacts on the caring unit, especially over their life course. In our application, the exosystem level moves from a limited welfare state with interventions focused on the individual care receiver to universal care policies, such as Social Security credits for caregivers [57]. Social Security disability is a current, individually focused response at this level, which is intended to support the care recipient. As noted earlier, adult caregivers lose more than USD 320,000 in wages and benefits over the course of a lifetime due to their provision of care [11]. Taking a chronosystem approach allows for a consideration of the ways in which youth caregiving will impact financial status in retirement. For example, if a young person is unable to engage in gainful employment as a young adult, they will be at an economic disadvantage later in life [58]. Current US public policy does not consider the ways in which caregiving youth are negatively financially impacted by early caregiving. Social Security caregiver credits would “… credit individuals serving as caregivers of dependent relatives with deemed wages for up to five years of such service.”, resulting in a strong financial foundation due to the provision of care.

Another example in the US related to the COVID-19 pandemic involved vaccination eligibility. If Marvin received the COVID-19 vaccine, which would have been the most common outcome, given the original CDC vaccination guidelines, he would have been protected from the virus [59]. However, an application of the SEM recognizes the value of vaccinating Tamara, in order to ensure the protection of the caring unit and maintain the functioning of the household [59]. If only Marvin was vaccinated, the overall risk of death of Marvin would be reduced, but the larger dynamic of the household and Marvin’s care depends on the well-being of Tamara. Although she is not at a high risk of death from the virus, her illness would significantly and adversely impact and household, as well as Marvin’s well-being and ability to continue living in the community. It becomes clear that all three members of this caring unit must be vaccinated in order to reap the benefits of vaccination to create a safe household at the microsystem level. In the US, young people were the last cohort eligible for vaccination during the initial rollout. Given the vulnerability of care receivers, a policy addressing the microsystem would provide greater safety for the care receiver and household unit. Below, we provide a further illustration of the application of the SEM to caregiving in Table 1.

Here, we utilize Bronfenbrenner’s model to demonstrate caregiving as it is typically approached and our application of the SEM to include both caregivers and care receivers. It is beyond the scope of this paper to enumerate all of the elements in Table 1. However, given our focus on the progression of life stages of caregiving youth, we direct our attention to the chronosystem to illustrate implications for research and practice. 

In Table 1, we attempt to address some of the criticisms of the SEM as cited above. Specifically, the oversimplification of systems that are largely underutilized in any meaningful way in developing policies and interventions is elaborated on here. For example, in Table 1, we draw attention to the need to address all members of the caregiving unit across each subsequent level in order to produce meaningful research and effective interventions. A valuable illustration is the underutilization of respite services, as mentioned above. Respite is a policy explicitly designed to support caregivers (exosystem). However, at the same time, it is implemented by a service agency (mesosystem) but delivered by an individual home care worker at the household (microsystem). Redesigning this intervention with the microsystem in mind would support implementation that would be more family- and household-centered. This reimagining of respite services would be more effective in offering the type of support that is currently intended, but not designed into current programs, which ignore the different contexts in which care takes place. A comprehensive reimagining of all current and future policies and interventions would benefit from similar applications of the SEM.

## 7. Conclusions

The majority of long-term care in the US takes place within the home by friends and family caregivers of all ages. While there has been progress in relation to research, policy, and practice related to caregiving youth, substantial gaps remain [60]. We propose a framework to address these gaps, integrating the SEM. Specifically, we have reconceptualized the individual level, around which subsequent spheres are organized, to point to the relational nature of care. Drawing attention and shifting from the individual to a relational unit of analysis has implications for research, policy, and practice. Utilizing this framework broadens the scope of who can be served by policy, focusing on collective needs. Current policy and practice neglect caregiving youth [10,60,61]. By focusing more inclusively on the actual dynamics of care taking place, policy will better capture people who otherwise fall through the cracks [61]. While it is beyond the scope of this paper to enumerate the particulars of each level, we make an effort to provide examples and set the stage for this inclusive way forward and the implications on research, policy, and practice.

Current research, policy, and practice overly focus on the individual and minimize the critical role that microsystem relationships of care play in the lives of care receivers, as well as paid and family caregivers across the life course. Their outcomes are contingent upon each other, but this is ignored when interventions are developed with only one individual in mind. The efficacy of interventions will continue to be limited because the context of care is relational. We assert that future research on caregiving must more systematically integrate caregivers and care receivers to fully contextualize the conditions of care, as well as consider how life stages impact the experiences of giving and receiving care, the practices and impacts of interventions, and the outcomes of all involved. Given that Bronfenbrenner’s model focused on child development, caregiving youth best illustrate the intertwined nature of care and the need to be vigilant about contextualizing the conditions of care across the systems. Moreover, the application of the SEM is valuable in relation to caregiving youth, not because they are vulnerable in comparison to adult caregivers, but because of the state of their development and the impact on care. We hope this application can be utilized in service to all caregivers and care receivers through an expanded and more integrated approach to research, policy, and practice.

## Figures and Tables

**Figure 1 ijerph-21-00119-f001:**
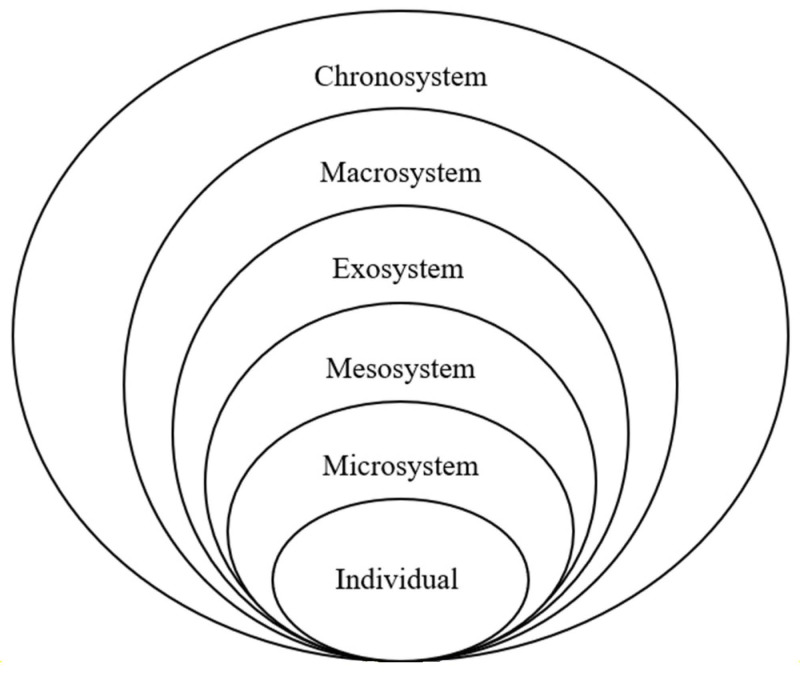
Bronfenbrenner’s ecological systems theory.

**Table 1 ijerph-21-00119-t001:** Re-envisioning caregiving within the social–ecological model.

	Bronfenbrenner	Current Conceptualization	SEM Application
Individual	Child (theory of development)	Care receiver	Caregiving unit (care receiver, family caregivers, paid caregivers)
Microsystem	Family, peers, schools, services, church	Family, home, delivery of services in the home (paid caregivers)	Reconfiguration of households and families of choice who fill roles in a variety of ways; paid care
Mesosystem	Neighborhood, social organizations, physical environment (urban, suburban, rural)	Same, with a focus on home and community-based service providers (i.e., home care agencies)	Universal and participatory family-centered design of services, which are flexible, inclusive, and responsive
Exosystem	Social services (organization of services), local politics, industry, mass media, government	Limited welfare state (SSD, SNAP (Social Security Disability and Supplemental Nutrition Assistance Program)); for-profit social services (i.e., respite); invisibility of care/limited coverage of care in media	Universal care policies, including paid family leave; workplace policies; Universal Basic Income (UBI); SS Caregiver Credit Act; state and federal policies
Macrosystem	Cultural and political ideologies; regulatory and constitutional frameworks	Personal responsibility (i.e., individualism, neoliberalism); care as women’s work	Collectivism; interdependence; gender equity in relation to caregiving
Chronosystem	Change over time: socio-historical conditions or patterns of events and transitions over a life course	Caregiving is present from onset through the life course with no accounting for different life stages	Responses to care are life course sensitive (caregiving youth: 8–18 years, young adult caregivers: 18–25)

## Data Availability

No new data were created in this study. Data sharing is not applicable to this article.

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
