# Peer review of "The Social Ecology of Caregiving: Applying the Social–Ecological Model across the Life Course"

_ijerph, 2024, doi:10.3390/ijerph21010119_

Round 1

Reviewer 1 Report

Comments and Suggestions for Authors

This paper sets out to propose a programme of work to address the perceived issue of the shortcomings in the provision of inter-generational care in particular that which sees the children of those needing care providing it. This is an important issue which the authors are to be congratulated for exploring and is arising due to the changing demographics of many developed countries and economic circumstances. Their work is timely and will have relevance to a wide audience 

Using what the authors have termed as an SEM (socio-ecological model) approach and a single case study to highlight the issues, the authors conclude that their analyses highlight the value of the SEM approach along with various issues which require further work that will help inform policy and the necessary interventions to address the current problems.

While as mentioned above the subject matter is of relevance and importance, there are currently  several issues which need to be addressed in order to merit publication. These include:

In their first (introduction) section, the authors state “we systematically apply the SEM as an effective framework…”. This is the central issue of their work. The authors need to break up this first section into a background which describes the issues and subsequently the shortcomings of the existing discussions and lines of research of caring. Indeed, the authors should provide a reference for their comment about the Olmstead Decision as many readers will be outside of the United States.

This provides the lead into the second issues: the justification for using SEM as the approach and the shortcomings of other arrangements, namely why SEM? Again, given the importance and potential value of their work, readers outside of sociology would benefit from a more precise justification for the adoption of SEM. 

The current section 2 (methods) needs work as it is more a case study that the authors have used. This may be an example to highlight how SEM can be used to add to the current information informing policy. This should be a new section 3.

Finally, the current conclusions require work. The main point made is that the ‘chronosystem’ is of importance. This may be the case but it is simply one element, indeed, the paper may be better titled to illustrate this point. As the authors have noted, this is part of a far wider piece of research and there is a sense that the current paper is trying to convey a larger presentation than is appropriate at this stage. 

Reviewer 2 Report

Comments and Suggestions for Authors

I have the following observations about this paper which seeks to be understand youth/young adult caregiving from an ecological perspective.

1. The introductory material on caregiving, while well written seems a bit disconnected from the paper's ecological focus. It should be shortened and reframed in light of the point beyond which the paper builds upon the work of Talley & Crews. I also think the authors should decide explicitly who they want to focus upon- youth, emerging adults, or young adults or all of the above. Them material on emerging adulthood, if that is their focus, needs to be better described as per its volitional nature. 

2. While a dyadic/ecological approach to youth/young/emerging adults is advantageous in many respects, issues of the bidirectionality of influence as well as synchrony/asynchrony are ignored, as is a family systems approach to caregiving. The elements of the SEM do not only interact, but their influence is reciprocal. Likewise, they may not work together in a synchronous manner-each of these ideas to include those underlying family systems' functioning has implications for the fit between elements of the caregiving dyad as well as other components of the Bronfenbrenner approach. Thus, a dialectical model may need to be integrated into the authors' approach, as does a more clearly defined developmental focus. 

3. The literature on caregivng where relevant to the present paper needs to go beyond public health-there are likely hundreds of studies that are relevant in the fields of social work, psychology, aging, and sociology. If the authors are going to incorporate Bronfenbrenner's work, which is not public health in origin, then they should extent that focus to caregiving research, i.e., the view of caregiving as a career by Aneshensel or the Pearlin antecedent-mediator-outcome model of caregiving. 

4. The figures add nothing to this paper-only one is needed to define Bronfenbrenner's models with examples. Table 1 is far more valuable.

5. I wonder whether the extensive autobiographic/autoethnography is necessary-if it is kept it needs to be annotated more adequately as a reflection of the SEM and the technique itself needs to be better explained and framed within the SEM.

6. While the paper is clearly written, it would benefit from an organizational overhaul so that salient points in both the introduction, methods/results and discussion can be presented so that they dovetail together more adequately, especially as it relates to the SEM as derived from Talley & Crews and Bronfenbrenner, to say nothing of the above conceptual additions. At present, the paper is overwritten and has a run-on quality of ideas to it-it needs to be shorter, more focused, and needs more structure, as reflected in more specific subheadings. The conclusions section does not seem to add much as well. How does the literature fall short? What are the parallels between Table 1 and implications for interventions, caregiving research, practice, policy?

Round 2

Reviewer 1 Report

Comments and Suggestions for Authors

The authors have undertaken a substantial review and have addressed the key issues in my first review. As such I am happy for the present work to go through to publication.

Reviewer 2 Report

Comments and Suggestions for Authors

While it was difficult to get a comprehensive picture of the new/revised paper given all of the changes, where an entirely new version minus the omissions would have been preferable, it appears from the authors' responses and what I could get from the revised paper that it is much improved.